# Insulin and Insulin Receptors in Adipose Tissue Development

**DOI:** 10.3390/ijms20030759

**Published:** 2019-02-11

**Authors:** Angelo Cignarelli, Valentina Annamaria Genchi, Sebastio Perrini, Annalisa Natalicchio, Luigi Laviola, Francesco Giorgino

**Affiliations:** Department of Emergency and Organ Transplantation, Section of Internal Medicine, Endocrinology, Andrology and Metabolic Diseases, University of Bari Aldo Moro, 70124 Bari, Italy; angelo.cignarelli@uniba.it (A.C.); valengenchi@gmail.com (V.A.G.); sebastio.perrini@uniba.it (S.P.); annalisa.natalicchio@uniba.it (A.N.); luigi.laviola@uniba.it (L.L.)

**Keywords:** insulin receptor, adipose tissue, adipocyte, receptor isoform

## Abstract

Insulin is a major endocrine hormone also involved in the regulation of energy and lipid metabolism via the activation of an intracellular signaling cascade involving the insulin receptor (INSR), insulin receptor substrate (IRS) proteins, phosphoinositol 3-kinase (PI3K) and protein kinase B (AKT). Specifically, insulin regulates several aspects of the development and function of adipose tissue and stimulates the differentiation program of adipose cells. Insulin can activate its responses in adipose tissue through two INSR splicing variants: INSR-A, which is predominantly expressed in mesenchymal and less-differentiated cells and mainly linked to cell proliferation, and INSR-B, which is more expressed in terminally differentiated cells and coupled to metabolic effects. Recent findings have revealed that different distributions of INSR and an altered INSR-A:INSR-B ratio may contribute to metabolic abnormalities during the onset of insulin resistance and the progression to type 2 diabetes. In this review, we discuss the role of insulin and the INSR in the development and endocrine activity of adipose tissue and the pharmacological implications for the management of obesity and type 2 diabetes.

## 1. Introduction

Adipose tissue (AT) is a critical regulator of energy balance and substrate metabolism, also through the production and secretion of several substances with endocrine or paracrine functions that are involved in energy homeostasis. An excessive amount of AT, particularly in the visceral depot, has been associated with the development of type 2 diabetes (T2D), premature atherosclerosis and cardiovascular disease. The biological features of AT from different sites could play a crucial role in the onset of metabolic derangements observed in overweight and obese subjects, and this may include the different sensitivity of specific AT depots to the action of insulin. Moreover, the contribution of different insulin receptor (INSR) splice variants to adipocyte development and function is not completely understood. Understanding the molecular basis of AT response to insulin is important for the pharmacotherapy of diabetes, also in relation to the peculiar effects of specific insulin analogues on fat expansion and weight gain.

## 2. Insulin Action and AT Metabolism

Insulin exerts a critical control on anabolic responses in AT by stimulating glucose and free fatty acid uptake, inhibiting lipolysis and stimulating de novo fatty acid synthesis in adipocytes (Figure 1). In addition, insulin regulates AT growth and differentiation, by enhancing the gene expression of various fat-specific transcription factors, including SREBP-1c and PPARγ [1].

One of the main actions of insulin is the regulation of carbon energy deposition, primarily through the control of glucose metabolization [2]. Insulin increases glucose uptake in adipocytes by regulating the intracellular localization of GLUT4 (the main glucose transporter involved in the insulin-regulated glucose transport) from the cytosol compartment to the plasma membrane [3]. Although only ~10% of the whole-body insulin-stimulated glucose uptake occurs in AT, the amount of AT GLUT4 represents a good marker of systemic insulin sensitivity [4]. Interestingly, glucose uptake displays significant differences between subcutaneous AT (SAT) and visceral AT (VAT) in humans, as assessed with a standard noninvasive ^18^FDG-PET technique, showing that VAT has higher rates of insulin-stimulated glucose uptake than SAT, both in lean and obese subjects [5]. In line with this concept, insulin signaling is more rapidly and prominently activated in VAT than SAT in humans [1,6].

Fat accumulation is determined by the balance between the synthesis and breakdown of triglycerides (TGs). Upon feeding, fatty acids accumulate in AT from two distinct sources: Circulating TGs and de novo lipogenesis (DNL) [7]. Circulating TGs are originally synthesized in the intestine or liver and packaged into chylomicrons or very low-density lipoproteins (VLDL), respectively. When these lipoproteins travel to AT, the TGs are hydrolyzed into non-esterified fatty acids (NEFA), following the action of the insulin-stimulated lipoprotein lipase (LPL) located within the vascular endothelium of AT [8]. Released NEFAs enter adipocytes through fatty acid transporters such as CD36 [9] and fatty acid transport protein-1 (FATP1) [10], whose translocation to the plasma membrane is also stimulated by insulin [11]. Meanwhile, insulin fosters adipocyte glucose uptake, which drives a small fraction of DNL, since the majority of this process is accounted for by the esterification of preformed fatty acids [12]. The first committed step in fatty acid synthesis is catalyzed by fatty acid synthase (FAS), a multifunctional cytosolic protein that primarily synthesizes palmitate, a 16-carbon saturated fatty acid [13]. FAS is positively regulated by insulin, mostly through transcriptional effects [14]. Interestingly, variations in FAS expression and enzyme activity have been implicated in insulin resistance (IR) and obesity in humans [15,16,17]. Fatty acids from these two sources are then esterified using glycerol 3-phosphate, derived from glucose, as a backbone to form TGs that are stored into lipid droplets [18]. Insulin also stimulates adipogenesis and lipogenesis through the induction of SREBP-1c and several other transcription factors involved in human adipocyte differentiation [19,20].

Another important action of insulin in fat cells is to limit lipolysis by inhibiting hormone-sensitive lipase (HSL) [21,22]. This action of insulin is indeed important, as insulin is the most potent antilipolytic hormone and acts rapidly in this regard. During exercise and starvation, the activation of β-adrenergic receptors induces lipolysis through the activation of cyclic adenosine monophosphate (cAMP) formation by adenylyl cyclase, a process mediated by the G protein Gs [23]. In contrast, during the fed state, insulin exerts a strong antilipolytic effect by inducing the AKT-mediated activation of phosphodiesterase 3B (PDE3B), which results in an increased rate of cAMP degradation [24].

### 2.1. Regulation of AT Mass

AT is formed at specific times and locations. Once formed, the tissue retains dynamicity, responding to homeostatic and external signals and being capable of a 15-fold expansion [25]. Adipose-derived stem cells (ASCs) are the principal components of AT involved in homeostatic maintenance and tissue development, and they are highly responsive to insulin. AT development and expansion occurs both through the differentiation of ASCs into adipocytes and through the storage of lipids, and insulin participates in these processes by stimulating both adipocyte hypertrophy (or lipogenesis) and hyperplasia (or adipogenesis). The level of activation of each of these two responses varies depending upon genetic background, modifier effects, diet, biological and hormonal milieu and the anatomical site of fat. Early studies identified insulin to be an inducer of preadipocyte differentiation [26], even though this effect was traditionally attributed more to the activation of the insulin-like growth factor-I receptor (IGF-IR) than INSR [27] with supraphysiological concentrations of insulin, also considering the relatively high expression of IGF-IR compared to INSR in preadipocytes [28]. Indeed, the INSR plays a fundamental role in stem cell commitment to adipogenesis during embryogenesis. In a mouse model of INSR knock-out, induced pluripotent stem cells subjected to a protocol of adipocyte differentiation yielded a poor differentiation as compared to the control cells in terms of lipid droplet accumulation and specific gene expression [29]. These findings are consistent with the report from Boucher et al., that the INSR has a crucial role in the control of AT development [30]. However, the effects of insulin on human adipocyte maturation seem to be prevalent in the late phases of adipogenesis. In both the early and intermediate phases of adipogenesis, the expression levels of adipogenic markers in human ASCs undergoing differentiation without insulin were comparable to those in ASCs differentiated with insulin, while in the late phase of differentiation, the contribution of insulin was found to be essential to achieve a completely functional adipocyte phenotype [19].

AT can expand from 2–3% to 60–70% of body weight in response to a positive energy balance [31]. In an elegant study, Spalding et al. demonstrated that the number of adipose cells in humans is established around puberty, and that there is an annual turnover of the cells in SAT of ~10% [32]. However, the turnover of adipose cells decreases in hypertrophic obesity [33]. The hypertrophic response involves pre-existing adipocytes that increase TG storage under insulin stimulation, achieving a two- to three-fold increase in their volume [34]. When there is a long-term exposure to a hypertrophic environment, adipocytes reduce the sensitivity to insulin, thus favoring AT dysfunction [35]. Since adipocytes cannot store further lipids under these conditions, new small adipocytes are formed from preadipocytes, especially in the VAT, and this may even be protective against metabolic impairment [36]. Indeed, several studies have observed that during caloric excess, the ASC compartment may be induced to proliferate and differentiate. For instance, dietary inputs can modulate these two biological endpoints in adults [37]. Therefore, the ability of mature adipocytes to accumulate lipids and the ability of ASCs from the stromal vascular fraction (SVF) to form new adipocytes are the key processes underlying the regenerative capacity of AT.

Moreover, adipose progenitors share a common origin with endothelial and perivascular cells [38,39], suggesting that adipogenesis, angiogenesis and vascular remodeling are mechanisms that are tightly and coordinately regulated in a paracrine/endocrine manner [40]. Under physiological conditions, both white adipose tissue (WAT) and brown adipose tissue (BAT) are hypervascularized, and the adipose vasculature displays functional plasticity to comply with the metabolic demands of adipocytes. Moreover, blood vessels not only supply nutrients and oxygen to nourish adipocytes, but they also serve as a cellular reservoir to provide adipose precursor and stem cells that control the AT mass and function. Thus, physiological AT remodeling also requires an adequate perfusion and subsequent delivery of nutrients into fat cells [41]. Indeed, insulin can affect the vascular endothelium of AT by inducing widespread vasodilatation and capillary recruitment without significant changes in the endothelial nitric oxide synthase (eNOS) in healthy individuals [42,43]. Furthermore, several studies have reported that insulin increased endothelial cell migration via activation of the PI3K-Akt-SREBP-1-Rac1 pathway [44], and enhanced new vessel formation via glycogen synthase kinase-beta3 and eNOS [45].

### 2.2. Insulin Effects on AT Endocrine Activity

In addition to regulating the release and storage of lipids, AT functions as a large endocrine organ that regulates several aspects of whole-body physiology through the release of hormones, lipids and cytokines (adipokines) [46]. Research over the last two decades has shown that AT releases molecules such as cytokines and other proinflammatory molecules (e.g., retinol binding protein 4 (RBP4), TNFα, IL-6 and IL-1β), which can promote AT and systemic inflammation, thereby antagonizing insulin action and favoring the development of cardiometabolic abnormalities, particularly in insulin-resistant obese subjects [47,48]. However, AT can also secrete molecules that are associated with enhanced insulin sensitivity, such as adiponectin and the recently discovered branched fatty acid esters of hydroxy fatty acids (FAHFAs) [49]. In spite of several reports that the levels of “metabolically favorable” molecules decrease with obesity and IR, few studies have investigated the impact of insulin on the AT secretion of adipokines and other molecules (Table 1). An early study using omics approaches in mature 3T3-L1 adipocytes showed that the in vitro administration of insulin induced the expression or secretion of a total of 27 proteins (mainly through post-translational modifications), including adipsin (a serine protease that stimulates glucose transport for triglyceride accumulation), secreted acidic cysteine-rich protein (SPARC) (involved in cell reorganization and angiogenesis), complement C3, collagen and other components of the extracellular matrix (ECM) [50]. Another similar study in 3T3-L1 adipocytes showed that insulin significantly increased about 60% of secreted proteins principally associated to ECM remodeling, such as fibronectin, thrombospondins, collagens, dystroglycan and tenascin [51].

Among the numerous adipokines with endocrine activity, including factors involved in fat storage and metabolism and eating behavior, some have been described to be regulated by insulin both in vivo and in vitro. Adiponectin/ACRP30 is an adipose-derived protein with insulin-sensitizing and anti-atherosclerotic properties. It is abundantly present in plasma (range: 5–30 μg/mL), but its adipose mRNA and circulating protein levels decrease in obesity and T2D [71]. Hyperinsulinemic euglycemic clamp studies in humans and monkeys have shown that plasma adiponectin/ACRP30 levels correlate significantly with whole-body insulin sensitivity [72]. Moreover, Halleux et al. demonstrated that the incubation of human VAT for 24 h with insulin enhanced mRNA levels of adiponectin. Nevertheless, AT may exert a negative effect on its own production of adiponectin/ACRP30 by releasing factors that destabilize its transcriptional and translational processes [52]. Accordingly, the secretion of adiponectin was generally higher in human VAT than SAT cells following insulin treatment and was negatively correlated with BMI [53]. Beyond its direct effect on adiponectin production, insulin indirectly ameliorates the biological response to adiponectin by inducing an increase of Adipo R1/R2 receptor expression, principally involved in fatty acid oxidation and glucose uptake, in both physiological and pathophysiological states such as fasting/refeeding, insulin deficiency and hyperinsulinemia, and this occurs via the activation of the insulin/PI3K/FOXO1 pathway [54].

In addition, insulin can regulate the WAT production of leptin, an adipokine known to control feeding behavior by activating anorexigenic neurons, as well as energy expenditure [56]. The rates of leptin biosynthesis are positively correlated with BMI and fat cell size, however, the expression of leptin is also affected by insulin, which chronically stimulates leptin storage by pre- and post-translational mechanisms [73], suggesting that the insulin-induced release of preformed leptin could contribute to circulating levels of this hormone. Moreover, in vivo and in vitro studies have demonstrated that insulin increases leptin release by human subcutaneous abdominal and mammary AT, as well as in rat epididymal AT and 3T3-F442A adipocytes [57,58,59,61]. Insulin appears to stimulate leptin production via a PI3K/PDE3B-dependent mechanism in rat adipocytes [60], and via PI3K/AKT and the activation of the transcription factors SREBP1, C/EBP-α and Sp1 in human and rodent AT [55].

Apelin is a newly identified fat-derived hormone which is strongly associated with obesity and hyperinsulinemia [74]. Insulin was found to enhance apelin expression, since in a diabetic mouse model, the lack of insulin production causes a large decrease in apelin expression in adipocytes [61]. Apelin secretion is increased during insulin-dependent adipocyte differentiation in both human and mouse cells via PI3K, PKC and MAPK activation [62].

Recently, chemerin has been identified as a novel insulin-induced adipokine, implicated in autocrine/paracrine signaling to adipocyte differentiation and lipolysis [75]. The circulating and AT (subcutaneous and visceral) levels of chemerin were increased under hyperinsulinaemic conditions in women with polycystic ovary syndrome (PCOS) [63], as well as in ex vivo experiments on SAT and VAT explants of these subjects [63].

A potent and robust insulin-induced upregulation of lipocalin-2, a novel protein involved in obesity and diabetes, was also shown in VAT explants, occurring via the activation of both PI3K and MAPK signaling [64].

On the other hand, insulin appears to negatively regulate two adipokines: resistin and omentin. Resistin is a member of the newly discovered cysteine-rich secretory protein family and has been associated with reduced systemic insulin sensitivity [76]. The secretion of resistin was suppressed in a mouse model of hyperinsulinaemia and during 3T3-L1 adipocyte differentiation by the activation of proteins that induce the degradation of its transcript via PI3K, ERK or p38 mitogen-activated protein kinase (MAPK) independent pathways [65,66,67,77]. Similar results have been reported for omentin, a cytokine principally secreted by VAT, whose physiological concentrations achieve 100–800 ng/mL in humans and are modified in several pathological situations, such as obesity and insulin resistance [78,79]. Recently however, omentin was shown to enhance insulin-mediated glucose transport with changes in basal glucose transport, indicating that it has no intrinsic insulin-mimetic activity but may increase insulin signaling via the IRS protein [80]. The secretion pattern of omentin was evaluated ex vivo after the administration of glucose and insulin to human VAT explants, as well as in vivo, under an insulin/glucose infusion in healthy individuals, and a significant decrease of its release was found in both settings [68].

Vaspin (visceral adipose tissue-derived serpin, serpinA12) is a member of the serine protease inhibitor family of serpins, whose expression shows the highest values when plasma insulin levels and obesity peak, while they are lower in the presence of diabetes [81]. Moreover, serum vaspin concentrations show diurnal fluctuations with a pre-prandial rise and a post-prandial fall, probably due to insulin level excursions, as also confirmed during an insulin tolerance test in healthy individuals with reduced vaspin serum concentrations shortly after insulin administration [69,70].

Altogether, these findings support the concept that nutritional status, and consequently insulin levels, directly affect the production and release of multiple adipokines, which in turn may regulate glucose homeostasis, insulin sensitivity and the energy balance. Therefore, deciphering the molecular mechanism underlying insulin-induced adipokine release could potentially lead to new therapeutic interventions against obesity and diabetes.

### 2.3. IR and AT Expansion: Chicken or Egg?

IR is a condition reflecting the reduction of insulin-mediated glucose uptake into the key insulin-sensitive tissues and is usually characterized by high circulating insulin levels, due to the compensatory enhancement of pancreatic insulin secretion [82]. This could lead one to believe that AT expansion could be fostered by peripheral IR, since high insulin levels may induce glucose uptake, lipogenesis and lipolysis inhibition in this tissue. On the other hand, dysfunctional AT develops as a result of its expansion, and this may favor IR. Interestingly, not all overweight/obese individuals have IR, and the extent of IR does not always correlate with the extent of AT expansion.

A major determinant of metabolic health is the ability of SAT to store excess fat rather than allowing its accumulation in ectopic sites, such as the liver (as in nonalcoholic fatty liver disease), muscle and heart, as well as in epicardial/pericardial and visceral AT [83]. Lipotoxicity has emerged as a key factor underlying the development of metabolic abnormalities, both in the presence of dysfunctional AT, as well as with the partial or complete absence of AT, as observed in lipodystrophies. Lipodystrophies represent genetic/acquired models of the incompetence of AT to respond to insulin by storing energy. The reduced ability of the SAT to store TGs results in increased lipolysis and the ectopic accumulation of fatty acids as TGs in the pancreas, muscle and liver, as well as in VAT, leading to the typical metabolic alterations [84]. Thus, it is important to recognize that AT has a “fat buffering” property, which is key to preserve a normal level of insulin sensitivity and appropriate metabolic responses. Moreover, AT is also responsible for generating ATP through fatty acid beta-oxidation in mitochondria, providing maintenance for a wide range of cellular processes, such as growth and differentiation [85].

The causative mechanisms mediating the development of IR in AT have been investigated less thoroughly than in the muscle and liver and appear to involve defects in multiple steps of insulin signaling downstream from the INSR [86]. Animal models of diabetes and obesity show an approximate 50% decrease in INSR kinase activity and binding affinity in AT [87], and diabetic individuals show an overt reduction of the INSR tyrosine kinase activity associated with decreased INSR protein content in adipocytes [88]. Furthermore, long-term exposure to a high-energy diet can induce IR in AT by causing a reduction in INSR content, probably through the disruption of the lipid bi-layer of adipocytes due to activation of peroxidation processes [89]. Recently, specific miRNAs have been indicated as possible mediators of insulin resistance in AT, such as miR-27b, which is upregulated in different in vitro and in vivo models of IR and directly suppresses INSR expression by targeting the 3’UTR of INSRs [90]. Moreover, deficiency of leptin secretion or action in mice with obesity and lipodystrophy, respectively, can affect insulin signaling, leading to chronic hyperinsulinaemia associated with hyperglycemia due to a down-regulation of IRS-2, a key mediator of insulin signaling [91]. The insulin-stimulated glucose transporter GLUT4 is another key mediator of insulin action in AT. The expression levels of GLUT4 are reduced in adipose cells from insulin-resistant obese and prediabetic subjects (i.e., high-risk individuals with first-degree T2D relatives), as well as in T2D subjects [92]. GLUT4 protein levels in AT are also a marker of whole-body insulin sensitivity, measured with the euglycemic clamp technique [4]. Finally, a decreased relative abundance of fatty acid binding protein 4 (FABP4) expression at the adipocyte plasma membrane has been reported in T2D obese subjects, and this may also contribute to the abnormalities in the storage of TGs [93]. All of these mechanisms (i.e., reduced insulin signaling, reduced glucose transport, reduced fatty acid transport) result in a failure of lipolysis suppression, impaired glucose uptake, and a consequent reduction of glycerol-3-phosphate production (required for fatty acid esterification [94]), as well as reduced fatty acid availability intracellularly, thus limiting the nutrient storage properties of AT and favoring the lipid overload in ectopic tissues [95]. Specifically, when glucose uptake and utilization by adipocytes are reduced, lipolysis will proceed unrestrained and circulating NEFA concentrations will increase [96], as in shown in the early days in obese nondiabetic subjects using ^14^C-palmitate in combination with the insulin clamp technique [97]. Excess deposition of fat as LC-fatty acyl CoAs, diacylglycerol and ceramide in the liver and muscles will cause IR in these tissues, and in the case of the β-cell, this will lead to impaired insulin secretion and β-cell failure. AT is one of the first tissues to respond to a nutritional overload, with such alterations impacting other tissues.

Interestingly, the severity of IR varies in relation to the different AT compartments, and visceral adiposity is strongly associated with whole-body IR, being responsible for a higher lipid load in the portal vein, compromising the hepatic glucose balance. Increased adipocyte size is associated with higher serum insulin concentrations, IR and an increased risk of developing T2D [98]. Indeed, severely obese individuals with a healthy metabolic profile have smaller adipocytes and increased circulating adiponectin levels than obese individuals with adverse metabolic features [99]. The correlation between adipocyte size and IR has led to confirm the “adipocyte overflow” hypothesis, since large hypertrophic adipocytes are no longer able to store further lipids, causing the spillover of fatty acids into ectopic sites, resulting in IR. On the other hand, reduced blood supply may represent another important factor for the impairment of in vivo insulin-mediated glucose uptake in AT [96,100], and inappropriate blood supply to SAT and VAT, as observed during exaggerate adipocyte enlargement, may contribute to the reduction of in vivo insulin-mediated glucose uptake. Indeed, vascular and endothelial dysfunction observed in obesity and insulin resistance largely results from the altered secretion of proinflammatory cytokines, the decreased release of adiponectin from AT and postprandial hyperglycemia [101]. The dysfunction of the endothelium is principally due to an imbalance between the production of vasodilator and vasoconstrictor molecules and is driven by the alteration of specific steps of insulin signaling, such as the PI3K/AKT pathway [43,102]. Moreover, insulin-mediated vasodilation may be differentially impaired in VAT compared to SAT in obese subjects, suggesting a role of altered vascular insulin signaling in promoting inappropriate rates of glucose uptake [96] and fat expansion [102]. Insulin resistance in endothelial cells can potentially play a role in glucose homeostasis through at least two mechanisms: The lack of a vasodilator effect of insulin and the reduction of the transendothelial delivery of insulin to its target tissues. Early studies in humans have shown that insulin-stimulated vasodilation could be a significant contributor to insulin-stimulated glucose uptake [103] by a mechanism involving eNOS, which can be impaired in people with obesity or type 2 diabetes [104]. Subsequent studies have suggested that an effect of insulin resulting in an increased number of perfused capillaries in a given tissue, a phenomenon known as capillary recruitment, may be more important for glucose tolerance than the insulin effect on total blood flow [105].

## 3. INSR in AT

Different studies in animal models of IR conducted using the tissue-specific gene-deletion of INSR have allowed for the elucidation of the physiological roles of insulin, as well as the mechanisms underlying the development of IR in specific tissues (Table 2). However, different results have been obtained according to the selectivity of INSR gene deletion (e.g., in whole fat or adipocytes) based on the vector utilized.

The depletion of INSR in both WAT and brown AT (BAT) in mice, obtained through the Cre-recombinase/aP2 system (FIRKO), resulted in an impairment of insulin-mediated glucose transport and suppression of lipolysis, reduced fat mass without insulin resistance or glucose intolerance and extended lifespan [82,106]. However, transgenic mice with selective ablation of INSR in GLUT4 expressing tissues (GIRKO) develop diabetes and insulin-resistant AT with morphological changes of BAT (i.e., increased lipid droplet size and striking disruption of the multilocular structure) and heterogeneity of WAT [107], suggesting that the impaired insulin-induced glucose disposal derived from INSR deficiency differently affects the sensitivity of specific insulin-sensitive tissues. Conversely, permanent abrogation of INSR expression in adipocytes (AIRKO) resulted in severe lipodystrophy, with metabolic abnormalities such as IR, altered glucose homeostasis, dyslipidemia and fatty liver disease, leading to decreased lifespan [108,109]. Furthermore, INSR is also critical in adipocyte survival, as observed in a murine model of inducible INSR inactivation in mature adipocytes that developed WAT and BAT loss, cold intolerance and metabolic syndrome, but was then able to restore AT dysfunction after 10–30 days through regeneration mechanisms [110]. Recently, Merry et al. reported that partial peripheral IR disruption (PerIRKO^+/−^) caused mildly improved whole-body insulin sensitivity with no effects on lifespan compared to the complete peripheral INSR disruption (PerIRKO^−/−^), which resulted in a diabetic phenotype with reduced lifespan [111]. These data clarify the key role of INSR for AT development and function, highlighting its impact on the maintenance of glucose homeostasis and insulin sensitivity.

In gestational diabetes, the mRNA levels of INSR were significantly reduced in both SAT and VAT, with a relevant drop of INSR protein content in VAT, displaying an inverse association with key maternal and neonatal anthropometric/metabolic outcomes [114]. Thus, the alterations of INSR expression in human AT could also be relevant for the development of metabolic complications in offspring if they occur at the time of pregnancy.

### 3.1. INSR Isoforms

The metabolic action of insulin in sensitive tissues is the result of the differential distribution of INSR isoforms and IGF-IR at the cell surface. Insulin and IGF-I play a synergistic role on several endpoints (i.e., differentiation, apoptosis, metabolism), and thus the extent of the elicited responses reflects the balance of INSR compared to IGF-IR [115]. As already reviewed, the human *INSR* gene maps on chromosome 19 and encodes two isoforms depending on the exclusion or inclusion of 12 amino acids in the C-terminal domain, respectively, by a post-transcriptional exon skipping process. The short isoform (INSR-A) is predominantly expressed in undifferentiated cells and contributes to prenatal development and tissue growth, whereas the expression of the long isoform (INSR-B) is enhanced in post-mitotic and differentiated cells and is largely responsible for the systemic metabolic action of insulin in adults [116]. The differential expression of INSR isoforms derives from a tight regulation of mRNA maturation by several splicing factors, such as heterogeneous nuclear ribonucleoprotein (hnRNP) F promoting INSR-B expression and hnRNP A1 promoting INSR-A expression, or at post-translational level with furin involved in INSR-A cleavage and PACE4 supporting INSR-B maturation [117,118]. These events are also affected by growth factors, including insulin itself [119]. Furthermore, both INSR isoforms are co-expressed in most cell types and can form homodimers (i.e., INSR-A/INSR-A and INSR-B/INSR-B) and heterodimers (i.e., INSR-A/INSR-B), based on the sorting of the two variants into lipid raft microdomains. The INSR-A/INSR-B heterodimers are able to recognize both insulin and IGF-II with a similar affinity as INSR-A/INSR-A [120]. However, the trafficking of INSR isoforms may be differentially regulated by specific ligands, and this could also affect downstream responses. For instance, in fibroblast-like cells overexpressing the INSR-A isoform, insulin stimulates INSR-A internalization and regulates mitogenic and metabolic responses differently than IGF-II [121,122]. Moreover, both INSR-A and INSR-B are able to readily complex with IGF-IR hemidimers, according to the relative abundance of each isoform [123,124]. The resulting hybrid receptors (HRs) mediate different biological responses on the basis of ligand affinity and downstream signaling [125].

Alterations in INSR splicing are associated with IR and T2D, even though the results are somewhat conflicting. In one study, the INSR-A:INSR-B ratio was found to be reduced in adipocytes from diabetic patients, and it was suggested that this change could contribute to IR since INSR-B represents the major “metabolic” isoform in insulin-sensitive tissues [126]. However, other studies did not show any significant alterations in the INSR-A:INSR-B ratio in various forms of IR [127]. A recent study showed that the weight loss induced by either bariatric intervention or very low-calorie diet in obese humans may modify the INSR-A:INSR-B ratio by increasing INSR-B in both SAT and VAT, this being associated with improvements in insulin sensitivity and a reduction of fasting insulin levels [128]. However, the role of the distinct INSR isoforms in the development and function of human AT has not yet been fully clarified.

### 3.2. INSR/IGF-IR Hybrids

Insulin and IGFs share a 40–80% homology and synergistically regulate several biological functions, such as cellular growth and differentiation, glucose and nutrient metabolism, and survival/apoptosis [129]. As already reviewed, three ligands (insulin, IGF-I and IGF-II) bind to their own specific receptors (i.e., INSR and IGF-IR), but they can also bind to HRs, resulting from assembling hemidimers of one INSR αβ subunit with one IGF-IR αβ subunit. The INSR and IGF-IR have a high degree of amino acid sequence homology (84% in the kinase domain and 100% in the ATP binding pocket [130]), and share a similar intracellular signaling mechanism that mediates mitogenic and metabolic responses, although to a different extent according to the specific receptor. Indeed, the presence of partial structure dissimilarities in the INSR and IGF-IR molecules produce different affinities and potencies for the shared ligands, such that the INSR has a high affinity for insulin, but can also recognize IGF-II with 10–50-fold lower affinity and IGF-I with 100–500-fold lower affinity. By contrast, IGF-II binds to the IGF-IR with a 10-fold lower affinity compared to IGF-I. The HRs behave mainly as IGF-IR, with a greater affinity for IGF-I and IGF-II than insulin [123]. The HRs were first identified in human placentae [131], but are basically ubiquitous. Their relative abundance depends on the extent of expression and localization of each receptor, and can be calculated as follows: HRs = 2√IR√IGF-IR [132]. An in vitro study carried out in differentiated 3T3-L1 adipocytes showed that an increase of INSR expression induced the formation of HRs with IGF-IR hemidimers, which were still capable of fully stimulating glucose transport in response to IGF-I through activation of the INSR β-subunit [133]. HRs may play an important role in receptor signaling in normal and pathological tissues, particularly in human cancers. Patients with T2D display an increased expression of HRs in AT compared to non-diabetic control patients [134], and this could contribute to reduced insulin action on glucose uptake and the inhibition of pro-inflammatory responses, since the IGF-IR could act as a negative regulator of insulin signaling, as shown in preadipocytes [135].

To date, information on the role of HRs in human AT and adipocyte differentiation is limited. In vitro studies have demonstrated that preadipocytes from SAT of lean subjects exhibit an equal level of INSR and IGF-IR, while the INSR/IGF-IR ratio increased 10-fold after the conversion into mature adipocytes due to an increase of INSR expression for metabolic responses [28]. Moreover, treatment of these preadipocytes with both insulin and IGFs resulted in high rates of proliferation and glucose accumulation, due to the partial agonism of insulin and IGF-I on their own and in cognate receptors [135].

Recent experimental evidence suggests that the absence of one receptor cannot compensate for the lack of the other, suggesting the synergism of the two receptors. Despite IGF-IR deletion in vivo not being essential for the growth and development of BAT in the presence of the INSR, its expression is crucial for the full function of BAT in terms of cold acclimation and maintaining an appropriate balance of death and survival for fetal brown adipocytes (Table 2) [112,113]. Insulin and IGF-I signaling are required for fat development, since disruption of both receptors in AT (FIGIRKO) causes a loss of WAT and BAT, failure of adipocytes to differentiate and impaired thermogenesis, even though the effects of a hypercaloric diet on weight gain and glucose intolerance were mitigated [30]. On the contrary, selective abrogation of both INSR and IGF-IR in BAT is predisposed to obesity, severe BAT atrophy with mitochondrial dysfunction, and IR in mice fed with an obesogenic diet [113]. Another study reported that the adipocyte-specific deletion of INSR and IGF-IR led to inability to maintain body temperature, lipodystrophy, severe diabetes and ectopic fat deposition compared to IGF-IR loss alone, with a modest effect on fat physiology [108]. The mechanisms underlying the effects of both INSR and IGF-IR deficiency on AT may have an epigenetic origin, as recently found in brown adipocyte precursors lacking both receptors, which show a drop in several maternally and paternally expressed imprinted genes and miRNAs in a stable and heritable manner [136].

## 4. Pharmacology of INSR in AT

### Insulin Analogues

Insulin is the essential life-saver therapy for people with type 1 diabetes (7–10% of all diabetics). However, it is also a gold standard treatment for approximately 20–25% of patients with T2D [137]. Over the last few decades, specific insulin analogs have been designed to improve metabolic outcomes in diabetic patients by minimizing glycemic excursions and the risk of hypoglycemia [138]. Insulin analogs can be classified as short-acting (AspB10, lispro, aspart and glulisine) and long-acting (glargine, detemir and degludec), according to their pharmacokinetics/pharmacodynamic. Short-acting analogs can be administered shortly before a meal, since they rapidly disassemble in the subcutaneous injection site and are easily absorbed in blood capillaries. In contrast, long-acting analogs are usually administered once daily and allow for a slow and continuous method of insulin delivery. AspB10 was the first insulin analog to be developed, showing an increased affinity for both the INSR and IGF-IR with a high carcinogenic risk, and it is thus not used in clinical practice [139]. The selective effects of the various insulin analogs on the INSR isoform have been investigated by Sciacca et al. using an in vitro model of overexpression of INSR-A or INSR-B. The level of INSR-A and INSR-B tyrosine phosphorylation after stimulation with short- or long-acting analogs were similar to that of human insulin, however, aspart and lispro, but not glulisine, induced a more rapid extracellular signal-regulated kinase (ERK) activation through INSR-A, whereas all three analogs stimulated a more prolonged AKT activation compared to insulin [140]. On the other hand, the long-acting analogs glargine and detemir appeared to have a low affinity for the INSR-A isoform with a longer dissociation rate and a higher mitogenic to metabolic ratio compared to native insulin [140,141].

Insulin glargine reportedly promotes the differentiation of both subcutaneous and visceral preadipocytes, as well as lipogenesis [142]. However, these in vitro findings are somewhat difficult to reconcile with clinical data showing that some diabetic patients treated with lispro and glargine may develop lipoatrophy with severe AT inflammation [143,144,145]. On the other hand, insulin detemir was shown to induce the *Pparg2* adipocyte master gene to a lesser extent compared to human insulin, resulting in attenuated effects on adipocyte differentiation and lipogenesis in human subcutaneous and visceral ASCs, in spite of a similar activation of proximal insulin signaling [19].

In addition to antibodies, small synthetic peptides have been discovered that behave as INSR ligands. For instance, an insulin mimetic peptide (S597) displayed hypoglycemic effects in vivo but a limited mitogenic response, as well as higher lipogenesis and glucose uptake in vitro compared to native insulin [146]. This single-chain peptide, despite binding equipotency, appeared to activate only the metabolic arm of insulin signaling (i.e., glycogen synthesis) in L6 skeletal muscle cells transfected with the human INSR-A [147]. Another ligand, S519, discovered by phage display, activates INSR with sub-nanomolar affinity and exhibits agonist activity on both glucose uptake and lipogenesis [148]. The effects of these compounds on INSR isoforms and downstream signaling remain elusive. Two INSR isoform-selective agonists were also designed by mutagenesis approaches exhibiting tissue-specific responses in a rat model: INS-A, with a strong effect on glycogen synthesis in muscle and low lipogenic activity in adipocytes and INS-B, capable of inducing a high level of lipogenesis in adipocytes and glycogen accumulation in hepatocytes [149].

## 5. Conclusions

The role of insulin in regulating AT development and function is fundamental. Insulin stimulates glucose and fatty acid transport and lipid synthesis and suppresses lipolysis [1]. Furthermore, insulin exerts a pivotal role in modulating the late stages of adipocyte differentiation by inducing the expression of PPARγ, the master gene for regulation of adipogenesis [19]. In addition, several molecules with paracrine and endocrine activities, such as leptin, adiponectin, chemerin, omentin and vaspin, as well as proteins involved in ECM remodeling, have an insulin-dependent regulation, and there is a complex interplay between the vascular network and the adipocytes. New synthetic and natural agonists of the INSR have been recently developed in order to improve metabolic outcomes with minor effects on mitogenesis. Therefore, further studies will be important to design and characterize specific ligands that selectively activate more metabolically favorable responses in AT to counteract its dysfunction in obesity and T2D.

## Figures and Tables

**Figure 1 ijms-20-00759-f001:**
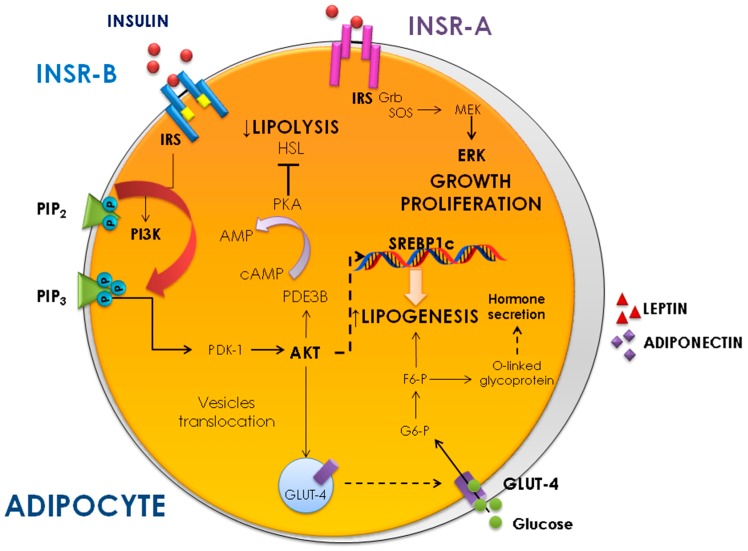
Insulin action in adipocytes. AMP, adenosine monophosphate; cAMP, cyclic adenosine monophosphate; AKT, protein kinase B; F6-P, fructose 6-phosphate; GLUT4, glucose transporter 4; G6-P, glucose 6-phosphate; Grb, growth factor receptor-bound protein; HSL, hormone-sensitive lipase; INSR-A, insulin receptor isoform A; INSR-B, insulin receptor isoform B; IRS, insulin receptor substrate; SOS, Son of sevenless protein; MEK, mitogen-activated protein kinase kinase; ERK, extracellular signal-regulated kinase; PDE3B, phosphodiesterase 3B; PIP_2_, phosphatidylinositol 4,5-bisphosphate; PIP_3_, phosphatidylinositol 3,4,5-triphosphate; PI3K, phosphoinositol 3-kinase; PDK-1, phosphoinositide dependent kinase 1; PKA, protein kinase A; SREBP1c, sterol regulatory element-binding protein 1.

**Table 1 ijms-20-00759-t001:** Insulin effects on adipose tissue (AT) endocrine activity.

References	Experimental System	Hormone/Cytokine	Function
[50,51]	In vitro 3T3-L1 adipocytes	Adipsin (↑) SPARC (↑) Complement C3 (↑) Collagen and extracellular matrix proteins (↑)	Stimulation of glucose transport Cellular reorganization, angiogenesis Immune response Tissue remodeling
[52,53]	Ex vivo Human VAT	Adiponectin (↑)	Insulin sensitizing action, Inhibition of atherosclerosis
[54]	In vivo Hyperinsulinaemia model	Adipo R1/R2 (↑)	FAs oxidation, glucose uptake
[55] [56,57] [58] [59] [60] [61]	In vivo Chronic Hyperinsulinaemia Ex vivo Human WAT Human SAT Rat VAT In vitro Rat adipocytes Human adipocytes	Leptin (↑)	Regulation of glucose and lipid metabolism and energy expenditure
[62]	In vitro Mouse and human adipocytes	Apelin (↑)	Angiogenesis, regulation of fluid homeostasis and energy metabolism
[63]	In vivo Human PCOS Ex vivo Human SAT/VAT	Chemerin (↑)	Regulation of lipolysis and adipocyte differentiation
[64]	Ex vivo Human VAT	Lipocalin-2 (↑)	Transport of small hydrophobic molecules (lipids, steroid hormones and retinoids)
[65,66,67]	In vitro Mouse adipocytes	Resistin (↓)	Reduction of systemic insulin sensitivity
[68]	Ex vivo Human VAT	Omentin (↓)	Metabolic regulation and anti-inflammatory effects
[69,70]	In vivo Healthy humans	Vaspin (↓)	Enhancement of systemic insulin sensitivity

↓, inhibition of expression/secretion; ↑, stimulation of expression/secretion; Adipo R1/R2, adiponectin receptors; FAs, free fatty acids; PCOS, polycystic ovary syndrome; SAT, subcutaneous adipose tissue; SPARC, secreted acidic cysteine-rich protein; VAT, visceral adipose tissue; WAT, white adipose tissue.

**Table 2 ijms-20-00759-t002:** In vivo effects of insulin receptor (INSR) and insulin-like growth factor-I receptor (IGF-IR) deletion in AT.

References	Mice Model	Vector	Location of Deletion	Effects
[82,106]	FIRKO	aP2/Cre	aP2 expressing cells	Fat mass (↓)
				Glucose transport (↓)
				Suppression of lipolysis
				Lifespan (↑)
[107]	GIRKO	GLUT4/Cre	GLUT4 expressing tissues	Diabetes
				Insulin-resistant AT
				Heterogeneous WAT
[108,109]	AIRKO	Adiponectin/Cre	Adiponectin expressing cells	Lipodystrophy
				IR
				IFG
				Dyslipidemia
				FLD
				Lifespan (↓)
[110]	AIRKO	Adiponectin/Cre + tamoxifen	Adiponectin expressing cells	WAT (↓)
				BAT (↓)
				Metabolic syndrome
				AT regeneration after 30 days
[111]	PerIRKO	Cre-ER + tamoxifen	Liver, WAT, skeletal muscle	Diabetes
				Lifespan (↓)
[112]	IGF-IRKO	Not available	Immortalized fetal brown adipocytes	Death (↑)
				Survival (↓)
[113]	F-IR/IGFRKO	UCP1/Cre	BAT	BAT atrophy
				Impaired thermogenesis
				Mitochondrial dysfunction
				Body fat mass (↑)
				IR (↑)
				Susceptibility to obesity
[30]	F-IR/IGFRKO	aP2/Cre	aP2 expressing cells	WAT (↓)
				BAT (↓)
				Impaired thermogenesis
				Energy expenditure (↑)
[108]	F-IR/IGFRKO	Adiponectin/Cre	Adiponectin expressing cells	WAT (↓) BAT (↓) Impaired thermogenesis Severe diabetes Ectopic lipid accumulation in liver, muscle and pancreatic islets

↓, decrease; ↑, increase; AT, adipose tissue; AIRKO, adipocyte-specific insulin receptor knockout; aP2, fatty acids binding protein; BAT, brown adipose tissue; Cre, recombinase cre; ER, estrogen receptor; FIRKO, fat-specific insulin receptor knockout; F-IR/IGFRKO, insulin receptor/IGF-IR double knockout; FLD, fatty liver disease; GIRKO, GLUT4-specific insulin receptor knockout; GLUT4, glucose transporter 4; IFG, impaired fasting glucose; IGF-IRKO, IGF-IR knockout; IR, insulin resistance; PerIRKO, peripheral insulin receptor knockout; WAT, white adipose tissue; UCP1, uncoupling protein 1.

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
