# Peer review of "Insulin and Insulin Receptors in Adipose Tissue Development"

_ijms, 2019, doi:10.3390/ijms20030759_

Round 1

Reviewer 1 Report

we do not have any comment for the authors

Author Response

We thank the Reviewer for his/her positive evaluation.

Reviewer 2 Report

In this manuscript, Cignarelli et al. reviewed available data regarding the role of insulin and its receptors in adipose tissue (AT) development.

Overall, the review is informative and exhaustive, even though some of its aspects need to be improved.

1.     The authors quickly mention (lines 126-130) the possible role of the vascular effects of insulin in AT remodeling. In fact, recent work underscores the importance of newly formed vascular networks in determining the plasticity of adipose tissue and its “healthy” expansion, in order to satisfy the increased storage capacity required in obesity. In this context, it would be interesting to discuss the involvement of insulin and its receptors in modulating and coordinating adipogenesis, angiogenesis and vascular remodeling in adipose tissue and, vice versa, the potential role of vascular insulin resistance in pathological fat pad expansion.

2.     Under the sub-heading “IR and AT expansion: chicken or egg?”, the authors correctly emphasize that different AT compartments, in particular, subcutaneous (SAT) vs. visceral (VAT) fat, may influence the severity of insulin resistance. In this regard, it would be important to better characterize the molecular mechanisms underlying insulin resistance in SAT and VAT. Again, the possible downregulation of insulin signaling in determining the inappropriate blood supply to SAT and VAT, involved in turn in obesity-related reduced glucose uptake (lines 280-283), should be discussed.

3.     Under the sub-heading “insulin analogues”, the authors indulge in a lengthy discussion of the pharmacological properties of different insulin analogues. This section, however, does not seem specifically related to the topic under review. The authors, therefore, should better focus their discussion in this regard and could safely shorted this section.

4.     The “Conclusions” section of the manuscript appears predominantly repetitious of concept already expressed before. The authors could more efficiently point out some unresolved issues and steer the needs of future research in the field.

Author Response

1.       “The authors quickly mention (lines 126-130) the possible role of the vascular effects of insulin in AT remodeling. In fact, recent work underscores the importance of newly formed vascular networks in determining the plasticity of adipose tissue and its “healthy” expansion, in order to satisfy the increased storage capacity required in obesity. In this context, it would be interesting to discuss the involvement of insulin and its receptors in modulating and coordinating adipogenesis, angiogenesis and vascular remodeling in adipose tissue and, vice versa, the potential role of vascular insulin resistance in pathological fat pad expansion.”

We thank the Reviewer for this suggestion. Accordingly, paragraphs on the role of insulin action on the adipose tissue vasculature and the consequence of vascular insulin resistance have been added on page 5, lines 127-139, and page 9, lines 295-311, respectively.

The additional following references have been added:

·        Gupta, R.K., Mepani, R.J., Kleiner, S., Lo, J.C., Khandekar, M.J., Cohen, P., Frontini, A., Bhowmick, D.C., Ye, L., Cinti, S., and Spiegelman, B.M. Zfp423 expression identifies committed preadipocytes and localizes to adipose endothelial and perivascular cells. Cell Metab. 2012, 15, 230–239. DOI:10.1016/jNaNet.2012.01.010

·        Wang, Q.A., Tao, C., Gupta, R.K., and Scherer, P.E. Tracking adipogenesis during white adipose tissue development, expansion and regeneration. Nat. Med. 2013, 19, 1338–44. DOI:10.1038/nm.3324

·        Escudero, C.A., Herlitz, K., Troncoso, F., Guevara, K., Acurio, J., Aguayo, C., Godoy, A.S., and González, M. Pro-angiogenic Role of insulin: from physiology to pathology. Front. Physiol. 2017, 8, 204. DOI:10.3389/fphys.2017.00204

·        Levy, B.I., Schiffrin, E.L., Mourad, J.-J., Agostini, D., Vicaut, E., Safar, M.E., and Struijker-Boudier, H.A.J. Impaired tissue perfusion: a pathology common to hypertension, obesity, and diabetes mellitus. Circulation. 2008, 118, 968–76. DOI:10.1161/CIRCULATIONAHA.107.763730

·        Frayn, K.N. and Karpe, F. Regulation of human subcutaneous adipose tissue blood flow. Int. J. Obes. 2014, 38, 1019–1026. DOI:10.1038/ijo.2013.200

·        Gogg, S., Smith, U., and Jansson, P.-A. Increased MAPK activation and impaired insulin signaling in subcutaneous microvascular endothelial cells in type 2 diabetes: the role of endothelin-1. Diabetes. 2009, 58, 2238–45. DOI:10.2337/db08-0961

·        Liu, Y., Petreaca, M., and Martins-Green, M. Cell and molecular mechanisms of insulin-induced angiogenesis. J. Cell. Mol. Med. 2009, 13, 4492–504. DOI:10.1111/j.1582-4934.2008.00555.x

·        Lassance, L., Miedl, H., Absenger, M., Diaz-Perez, F., Lang, U., Desoye, G., and Hiden, U. Hyperinsulinemia stimulates angiogenesis of human fetoplacental endothelial cells: a possible role of insulin in placental hypervascularization in diabetes mellitus. J. Clin. Endocrinol. Metab. 2013, 98, E1438-47. DOI:10.1210/jc.2013-1210

2.       “Under the sub-heading “IR and AT expansion: chicken or egg?”, the authors correctly emphasize that different AT compartments, in particular subcutaneous (SAT) vs. visceral (VAT) fat, may influence the severity of insulin resistance. In this regard, it would be important to better characterize the molecular mechanisms underlying insulin resistance in SAT and VAT. Again, the possible downregulation of insulin signaling in determining the inappropriate blood supply to SAT and VAT, involved in turn in obesity-related reduced glucose uptake (lines 280-283), should be discussed.”

We thank the Reviewer for this suggestion. Accordingly, a paragraph has been added on page 9, lines 295-311, as already noted. However, the literature on the specific insulin signaling abnormalities in SAT and VAT on insulin signaling individuals are limited.

We have included additional references to support these conclusions:

·        Ferrannini, E., Iozzo, P., Virtanen, K.A., Honka, M.-J., Bucci, M., and Nuutila, P. Adipose tissue and skeletal muscle insulin-mediated glucose uptake in insulin resistance: role of blood flow and diabetes. Am. J. Clin. Nutr. 2018, 108, 749–758. DOI:10.1093/ajcn/nqy162

·        Rask-Madsen, C. and King, G.L. Mechanisms of Disease: endothelial dysfunction in insulin resistance and diabetes. Nat. Clin. Pract. Endocrinol. Metab. 2007, 3, 46–56. DOI:10.1038/ncpendmet0366

·        Farb, M.G., Karki, S., Park, S.-Y., Saggese, S.M., Carmine, B., Hess, D.T., Apovian, C., Fetterman, J.L., Bretón-Romero, R., Hamburg, N.M., Fuster, J.J., Zuriaga, M.A., Walsh, K., and Gokce, N. WNT5A-JNK regulation of vascular insulin resistance in human obesity. Vasc. Med. 2016, 21, 489–496. DOI:10.1177/1358863X16666693

·        Laakso, M., Edelman, S. V, Brechtel, G., and Baron, A.D. Decreased effect of insulin to stimulate skeletal muscle blood flow in obese man. A novel mechanism for insulin resistance. J. Clin. Invest. 1990, 85, 1844–52. DOI:10.1172/JCI114644

·        Clark, M.G., Wallis, M.G., Barrett, E.J., Vincent, M.A., Richards, S.M., Clerk, L.H., and Rattigan, S. Blood flow and muscle metabolism: a focus on insulin action. Am. J. Physiol. Metab. 2003, 284, E241–E258. DOI:10.1152/ajpendo.00408.2002

·        Rask-Madsen, C. and Kahn, C.R. Tissue–specific insulin signaling, metabolic syndrome, and cardiovascular disease. Arterioscler. Thromb. Vasc. Biol. 2012, 32, 2052–2059. DOI:10.1161/ATVBAHA.111.241919

3.       “Under the sub-heading “insulin analogues”, the authors indulge in a lengthy discussion of the pharmacological properties of different insulin analogues. This section, however, does not seem specifically related to the topic under review. The authors, therefore, should better focus their discussion in this regard and could safely shorted this section.”

According to the Reviewer’s suggestion, this text on pages 12-13 has been modified to reduce this section.

4.       “The “Conclusions” section of the manuscript appears predominantly repetitious of concept already expressed before. The authors could more efficiently point out some unresolved issues and steer the needs of future research in the field.”

The Conclusions section has been rewritten accordingly.